# Implementation of Healthy Men Healthy Communities: A Health Promotion and Gender-Based Violence Prevention Program for Male South Sudanese Refugees in Uganda

**DOI:** 10.3390/healthcare12020147

**Published:** 2024-01-09

**Authors:** Ruth Zielinski, Daniel Kuir Ajak, Nora Drummond, HaEun Lee

**Affiliations:** 1School of Nursing, University of Michigan, Ann Arbor, MI 48109, USA; noradrum@umich.edu (N.D.); haeunlee@umich.edu (H.L.); 2South Sudan Leadership and Community Development, Grand Rapids, MI 49501, USA; dkuirajak@gmail.com

**Keywords:** male group, men’s health promotion, refugee settlements, gender-based violence prevention, sub-Saharan Africa

## Abstract

Men living in refugee settings are often exposed to violence, poverty, and social instability, which impacts physical and mental health and increases the risk of perpetrating sexual and gender-based violence. Healthy Men Healthy Communities was developed as a male-led health promotion program to address men’s physical and mental health and their role in creating healthy relationships and families. Three community leaders from the settlements were trained to facilitate the program, which was implemented among six groups consisting of twelve men in each group. Pre/post surveys and feedback were collected among the facilitators and participants. Facilitators suggested culturally appropriate ways to present physical activities as a stress reduction technique and the importance of spacing out births. The small group setting facilitated open conversations on topics such as birth spacing and healthy partner communication. Participants experienced an increase in knowledge and confidence in practicing the program content, such as stress-reduction techniques and healthy communication strategies. Participants recommended additional topics such as fertility and sexually transmitted infections. The Healthy Men Healthy Communities program has the potential for wider implementation among male South Sudanese refugees to promote their health as well as the health of their families.

## 1. Introduction

People in refugee settlements are highly vulnerable to poverty, trauma, communicable diseases, mental illness, and sexual and gender-based violence (SGBV) [1,2,3]. Correlations between male war experiences with poor mental health, unhealthy behaviors such as smoking and drinking, and engaging in SGBV are well established in the literature [4]. Additionally, most economic interventions in settlements target women specifically, which may unintentionally cause the men in refugee settings to experience a loss of their societal role within the family and community, which may further potentiate SGBV [1]. 

Uganda is one of the top refugee-hosting countries in Africa, with approximately 1.5 million refugees and asylum seekers residing there [5]. Currently, over 900,000 refugees are from South Sudan, with a continued influx of refugees due to the ongoing conflict in their home country [3]. Recent reports indicate a rise in SGBV among refugee settlements in Uganda, with the most predominant form being Intimate Partner Violence (IPV) [3]. Uganda’s Refugee Response Plan (2018) emphasizes the importance of engaging both men and boys in the development and implementation of programs aimed at eliminating SGBV [3]. Previous programs aimed at decreasing SGBV have successfully utilized community leaders to facilitate group education interventions [6]. However, missing from these interventions is content specific to men’s health and the impact of experiences such as war, poverty, and instability on male health and the rates of SGBV. 

## 2. Methods Section

Prior to this study, we conducted a needs assessment through interviews with male community leaders and local healthcare and nongovernmental organization members in two of Uganda’s refugee settlements. This assessment aimed to better understand the health and wellness needs of men in the community. Community leaders shared their belief that a lack of employment and means of caring for their family led to depression, alcohol use, and gender-based violence. Based on these findings, we developed a Healthy Men Healthy Communities program as a men-led, small-group format health program for men. 

### 2.1. Setting

Most of the nearly 1 million South Sudanese refugees living in Uganda reside in the north of the country in settlements close to the border of their home country [5]. The implementation of the program was conducted in two refugee settlements, one housing approximately 5000 people and the other 10,000 people. These sites were selected as community relationships had been established, and the community leaders had requested this program to be offered.

### 2.2. Curriculum Development

The Healthy Men Healthy Communities curriculum adopted a Home-Based Life-Saving Skills (HBLSSs) format for education. Developed by the American College of Nurse-Midwives, HBLSS is a community-based curriculum that is designed for learners who have low literacy and is implemented in a small-group, interactive format. HBLSS is used to educate traditional birth attendants and community members in limited resource settings utilizing storytelling, large picture cards, and group discussions [7]. The methodology of HBLSS was chosen as it had been implemented successfully with women in the same refugee settlements. The Healthy Men Healthy Communities curriculum retained the HBLSS format of training the trainer, small community meetings with participatory facilitation, storytelling, and picture cards (Figure 1). An interactive picture-based group discussion format, similar to HBLSS, was used in an effort to encourage participants to discuss their physical and mental health, practice stress reduction techniques, and role-play healthy partner communication in an all-male group setting. Existing evidence points to the importance of involving community leaders in engaging men in health promotion and SGBV prevention strategies [8]. To this end, the train-the-trainer approach was used, where facilitator training was conducted by male community leaders with other males. The community leaders then went on to facilitate three program meetings with small groups in the wider community.

### 2.3. Ethics Approval and Consent to Participate

This project was reviewed by the University of Michigan Institutional Review Board and determined to be non-regulated. No identifiers such as settlement names, locations, or direct participant quotes are included.

### 2.4. Curriculum Content

Curriculum topics were chosen by the leaders based on the interests expressed by men in the community. Health-promoting topics specific to men, such as a healthy lifestyle, stress reduction techniques, and healthy partner communication, were included. Materials from Safe Dates, an evidence-based curriculum for dating violence prevention, and Stepping Stones South Africa, a type of group-based gender transformative intervention on sex, alcohol use, and sexual and physical IPV, were also included in the intervention [9,10,11].

The curriculum consisted of three meetings. During the first meeting, the group learned about men’s general health, that health is not simply the absence of sickness but involves their bodies, minds, and relationships with their family and community. Information and opportunities for discussion about female health, such as the importance of birth spacing for the mother and children’s health, are included. During the second meeting, healthy strategies to reduce stress, such as walking away from the situation, taking deep breaths, participating in physical activity, and talking to a friend, were introduced, discussed, and practiced. In the final meeting, healthy communication strategies, such as the use of ‘I’ statements, were presented, discussed, and practiced.

### 2.5. Evaluation

After the facilitators completed the training in program implementation, they were asked to discuss their initial reaction to the curriculum, including the program’s potential value and possible challenges in its implementation. Detailed notes were taken during the facilitator training and during the feedback session following the training. Following program implementation, the 6 groups debriefed immediately after participating in the third and final meeting. However, detailed field notes were only available for one of the debriefing sessions with 12 participants due to the onset of the global pandemic. These participants also completed the survey. A short survey was given to these participants prior to the first meeting and following the last meeting. While the initial intent was to conduct the pre/post survey with all participants, travel to collect data was not possible due to the onset of the global pandemic. The pre/post survey included 7 questions aimed at assessing the difference in knowledge and awareness after the intervention but was not designed with the power to determine the significance of the difference.

## 3. Implementation

### 3.1. Facilitator Training

Three male community leaders who volunteered as facilitators completed the initial one-day train-the-trainer program. All three men had extensive experience in leading community education programs and were fluent in both English and the language spoken by the South Sudanese men in the settlements. English is the official language of South Sudan, and education is primarily provided in English, so the training materials were printed in English. However, most South Sudanese in these two settlements speak an indigenous language, so to be inclusive of diverse educational backgrounds, the indigenous language was utilized during the training in the settlement. 

The Healthy Men Healthy Community program includes a step-by-step facilitator’s guide with graphic images that are culturally appropriate and formatted to be displayed during education. A teach-back method was used whereby, in the morning, the facilitators were given the training and, in the afternoon, each “taught back” one of the sessions of the curriculum. 

### 3.2. Healthy Men Healthy Community Group Meetings

Each of the three facilitators led two Healthy Men Healthy Communities groups that met weekly for three weeks. The program participants were invited via word of mouth since we wanted the participants to invite people who were comfortable discussing potentially sensitive topics within the intimate group setting. The groups were limited to 12 participants due to the discussion-based and interactive nature of the curriculum. The meetings each lasted 1–2 h, with ample time to discuss the topics. Facilitators introduced ideas using short vignettes that included pictures, which were then discussed by the group. Participants had time to role-play ideas, such as stress reduction and conflict resolution techniques. An agreement would then be reached as to whether the technique was one that would be useful in their context and setting.

## 4. Results

### 4.1. Facilitator Debrief

Following the training session, the facilitators were asked to openly reflect and discuss the program format as well as the content of the curriculum. The debrief was guided using open-ended questions. Facilitators expressed confidence in implementing the program and felt the practice time during the training was particularly helpful. Overall, the facilitators found the content on healthy relationships, family planning, and healthy stress reduction techniques to be important in addressing issues that can arise due to geographic and economic instability and high levels of mental health distress associated with life as a refugee. Each topic was evaluated individually during the debriefing session. 

#### 4.1.1. Communication

The facilitators recognized that patriarchal gender norms prevalent in South Sudanese communities often made healthy partner communication challenging. One facilitator reflected that the cultural expectation for a Dinka man is to avoid displaying affection and giving public attention to their wives. The phrase “the wife has taken you” is said of men who are outwardly affectionate to their wives, such as by holding hands, sitting next to one another, or walking side by side in public settings. Being “taken by your wife” is negatively viewed by the community because such an explicit show of affection indicates that the man is prioritizing his wife and her needs rather than the greater needs of the community. One of the facilitators commented that if a man is excessively focused on his wife, it is viewed as a threat to the community. 

Facilitators described how men are often physically distant from their wives and children, either because of work or because their wives often stay with their mothers during pregnancy, childbirth, and the postpartum period. There is a saying in their indigenous language, “men don’t hear the cry of the baby”, which suggests that the social norm is that men’s involvement in child rearing is minimal. Facilitators believe the physical distance of men contributes to men’s challenges in being emotionally present and supportive of their wives and children. They felt that fathers might not see the suffering of children as acutely as the mothers would.

The facilitators acknowledged that, as a community, they need to push back against these cultural norms and utilize the curriculum’s content on communication skills to foster healthy relationships with wives. Participating in childcare, which is not a standardized practice for South Sudanese men, and showing tenderness and love in family interactions were also mentioned as methods for raising a healthy family.

#### 4.1.2. Birth Spacing

The facilitators also mentioned that the idea of spacing out children and having fewer children is a new but important idea for their community. Facilitators discussed how the concept of family planning and spacing out birth aligns well with the expectations for men to be providers for the family. Men are expected to provide for their family’s needs, such as food, housing, healthcare, and education, and the facilitators recognized that having fewer children and spacing them 3 to 5 years apart would allow men to better provide [12]. 

#### 4.1.3. Physical Exercise and Stress Reduction

The Healthy Men Healthy Communities curriculum included the physical and mental benefits of physical exercise. However, physical activities such as running, playing ball, and dancing are often viewed to be inappropriate for grown men. Facilitators shared the concept that running is associated with danger and people run only in emergency situations, such as being chased by a wild animal. Hence, men are not encouraged to run since it insinuates danger and could provoke fear and anxiety in the community. Furthermore, while children and youth often partake in soccer and dancing, the community would view that a man is not taking the plight of the refugee community seriously if he also partook in such activities. Facilitators discussed that with serious problems facing their community and South Sudan at large, it was challenging to be seen as enjoying oneself. However, during the discussion, facilitators explored strategies such as starting a recreational soccer league where a group of men could participate together and collectively experience the benefits of physical activity.

### 4.2. Post Program Implementation Debrief

#### 4.2.1. Content

Following the final meeting, one group participated in a debriefing session. Participants collectively reflected on the curriculum’s content, potential value, and areas for improvement. The participants found the most value in the stress reduction techniques, feeling that stress was a significant issue for them and some of the techniques were new to them. For example, one of the strategies taught that was new to participants was deep breathing when feeling angry. Participants enjoyed the time to practice what they learned such as communicating in a calm and constructive way. While all these techniques were interesting to the participants, they felt that incorporating the exercise may not be beneficial in reducing stress in their context. Participants felt that healthy communication content was important and informative, and they were eager to implement the strategies they learned. Regarding a healthy lifestyle, the participants discussed difficulties with implementing some of the content given the food, employment, and security challenges they faced in the settlements. However, they felt that the content should remain in the curriculum since it offered an opportunity for discussion and problem solving. 

#### 4.2.2. Format

The participants found that the small group setting was helpful for them to fully engage in the program, and discuss various content, and participate in role-play. One of the participants mentioned that “the group is very small, so everyone was able to participate and become knowledgeable”. Additionally, they wished the program was longer than the three-meeting format and suggested having it as a regular, ongoing meeting among men. Ultimately, they wished that they could teach and become good examples for younger men in the community. “Men are willing to take up the role of guiding youth or adolescents if they acquire knowledge”.

#### 4.2.3. Suggested Additional Topics

The participants recommended additional topics to be covered, such as men’s and women’s reproductive health, especially in terms of fertility. Participants expressed a desire to better plan for their family’s optimal health in terms of spacing out births and various long-term and short-term contraceptive methods. Furthermore, they wished to learn about other health topics, such as HIV/AIDS and stigmas related to the disease. Participants shared the concept that there was a stigma associated with HIV/AIDS in the community, which they believed led to an early death when those infected did not seek testing or treatment. When people were found to be HIV-positive, they were often treated badly or excommunicated, which often led to alcohol abuse or even suicide. 

### 4.3. Pre and Post Assessment Survey

There was an increase in all seven questions of the survey regarding men’s knowledge, acceptability, and confidence about the program content (Table 1). Compared to before participating in the Healthy Men Healthy Communities program, men showed a 20% increase in knowledge of the harmful effects of tobacco. Furthermore, there was a 9.1% increase in participant’s knowledge in understanding that “being healthy includes physical, mental, and social wellbeing”, “physical activity, such as playing soccer, can reduce feelings of anger”, and “it is healthy for both mothers and babies to allow at least three years between the birth of a baby and the next birth.” There was a 17.65% increase in men’s confidence in techniques to reduce feelings of stress and anger and a 15.8% increase in their confidence in communicating the use of “I” statements to avoid blaming their wives during conversations. Lastly, there was a 9.52% increase in men’s acceptability of the overall program goal: that it is important to learn how to be a healthy man and find ways to reduce stress.

## 5. Discussion

The facilitators’ responses, as well as the participant’s feedback and pre/post survey, suggest that the Healthy Men Healthy Communities is a promising program for implementation in this community. The facilitator discussion revealed that there are cultural barriers to practicing healthy communication with wives and participating in physical activities to relieve stress. However, considering the benefits of healthy communication strategies and physical activities, the facilitators felt that this content could be further culturally adapted within the curriculum. Considering how men in post-conflict settings face significant trauma, tension, and a sense of disempowerment, healthy coping strategies, as well as communication strategies with their partners, are essential [13]. Similar group-based interventions in war-affected regions of Côte d’Ivoire showed significantly positive results in men’s reported behaviors related to IPV [14].

Both the facilitators and the program participants reinforced the importance of topics such as family planning and a desire to learn more about similar reproductive health-related topics. A literature review that examined the factors influencing family planning in crisis-affected areas of sub-Saharan Africa found that male influence is one of the strongest factors in women’s decisions to utilize family planning services [15]. Hence, there is a critical need to better involve and educate men about family planning. Additionally, considering the international call for better paternal involvement in antenatal care to improve maternal and newborn outcomes, the further addition of content regarding male and female reproductive health, as well as healthy pregnancy and childbirth, could be beneficial [16]. The discussion-based small-group format of the program may further provide an excellent platform for male-group antenatal care, which has been understudied as a potential intervention to improve male involvement [16].

Facilitators and program participants unanimously expressed an interest in continuing the Healthy Men Healthy Community program and the need to add additional topics to the curriculum. When developing the curriculum, our team chose to include less-sensitive topics such as healthy communication during conflict as the starting point rather than beginning with topics such as contraceptive methods and SGBV. However, we found that both facilitators and participants desired more direct and specific content related to not only SGBV but also HIV, gender roles, healthy pregnancy, and childbirth. The next steps include adding additional content based on this feedback and implementing and evaluating this in other settings. 

Lastly, despite massive global gender disparities, policies and programs instituted by most national governments consistently fail to address men’s burden of ill health through male-centered strategies [6]. The Healthy Men Healthy Communities program aims to address not only men’s physical but also mental and social health to ultimately promote men, women, family, and community health in vulnerable settings such as refugee settlements.

### 5.1. Limitations

Overall, this study finding should be interpreted cautiously since all participants were male South Sudanese refugees, and the feedback and survey were collected from a limited number of participants. The feedback collected may be prone to social desirability bias since the facilitators who implemented the program were also part of the discussion, and participants may have overstated the benefits of the program. Given the small sample size, the survey was limited to descriptive statistics and percentages. Because the post-survey was conducted immediately after the program, the long-term outcomes related to improvements in knowledge and confidence are unknown. Finally, facilitator training was implemented immediately prior to the onset of the pandemic. Fortunately, COVID-19 did not severely impact northern Uganda, and the meetings, as well as the feedback and survey collection, were implemented. However, a more thorough in-person follow-up for non-refugee collaborators is limited. 

### 5.2. Conclusions

To achieve gender equality and address violence against women, men indisputably need to be part of the solution. In refugee settlements, men face a significantly higher risk of various physical and mental illnesses, which may contribute to conflicts within couples and families. The facilitator feedback, pre/post assessment survey, and participant feedback suggest that the Healthy Men Healthy Communities program may improve men’s knowledge and confidence in stress reduction techniques and healthy partner communication strategies. Furthermore, both facilitators and the participants expressed a strong desire to attend additional sessions with additional topics related to male and female reproductive health, healthy pregnancy and childbirth, and other health promotion and disease prevention content. Additional implementation and evaluation both within the South Sudanese refugee setting as well as in other communities is needed. Overall, this program has the potential to empower men as individual agents of change in their families and communities. 

## Figures and Tables

**Figure 1 healthcare-12-00147-f001:**
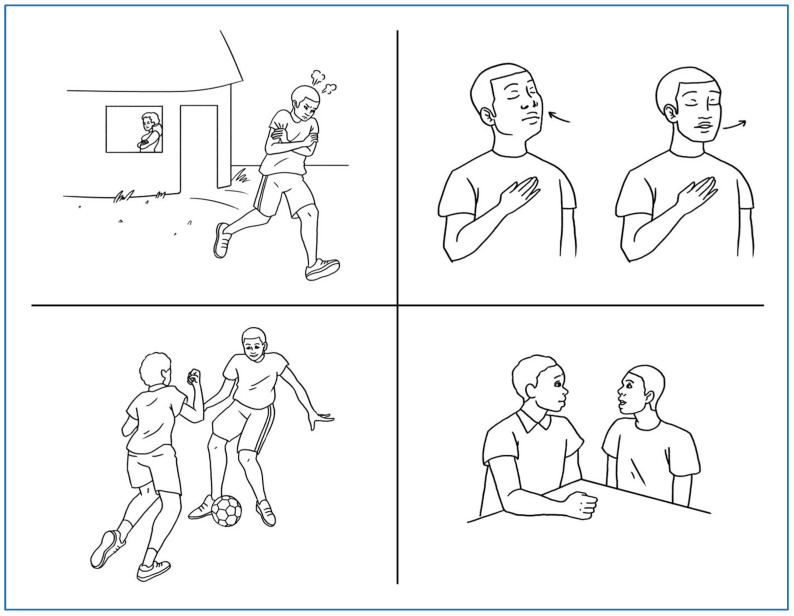
Example from Healthy Men Healthy Communities curriculum—Ways to reduce feelings of stress, frustration and anger.

**Table 1 healthcare-12-00147-t001:** Percent change in participants’ knowledge questions in pre/post test.

Knowledge Questions	% Change
1. Being healthy includes physical, mental and social well-being.	↑ 9.1%
2. One of the ways to be a healthy man is to avoid tobacco.	↑ 20%
3. Physical activity, such as playing football, can reduce feelings of anger.	↑ 9.1%
4. It is healthy for both mothers and babies to allow at least 2 years between the birth of a baby and the next pregnancy.	9.1%
Confidence Questions	% change
5. I am confident about techniques to reduce feelings of stress and anger.	↑ 17.65%
6. I am confident about communicating using statements that begin with “I” to avoid blaming the other person.	↑ 15.8%
Acceptability Question	% change
7. It is important to learn how to be a healthy man and find ways to reduce stress.	↑ 9.52%

## Data Availability

For data or curriculum requests, please email the corresponding author at ruthcnm@umich.edu.

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
