# Peer review of "Implementation of Healthy Men Healthy Communities: A Health Promotion and Gender-Based Violence Prevention Program for Male South Sudanese Refugees in Uganda"

_healthcare, 2024, doi:10.3390/healthcare12020147_

Round 1

Reviewer 1 Report

Comments and Suggestions for Authors

Hello, 

Thank you for the opportunity to review this paper. I have a few minor comments for your team.

2.1 Setting-How were these 2 refugee camps selected? Can you provide brief information about how you ended up in those 2 camps?

(There are two 2.1 sections). For the second 2.1 Curriculum Development, it's a little confusing about how the topics were picked to be presented. I understand that HBLSS was used as a guide for the actual training, but how did you choose the specific curriculum in the intervention?

4. Results-Can you clarify why you only did the pre/post survey with one of the six groups? Also, why only one group debrief? It seems like a huge missed opportunity for data. Although this is recognized in the limitation section, I think the authors should specifically address why they made this choice.

5. Discussion-I would like to see more about the next steps for this project and what the next steps will be? 

Author Response

Reviewer Comments

How and Where It was Addressed

Reviewer 1

2.1 Setting-How were these 2 refugee camps selected? Can you provide brief information about how you

ended up in those 2 camps?

Line 59-61 These sites were selected as community relationships had been established and the community leaders had requested this program be offered.

(There are two 2.1 sections). For the second 2.1 Curriculum Development, is a little confusing about how

the topics were picked to be presented. I understand that HBLSS was used as a guide for the actual

training, but how did you choose the specific curriculum in the intervention?

Line 62  Thank you for noticing this error, it has been corrected.

Line 68-70 The methodology of HBLSS was chosen as it been implemented successfully with women in the same refugee settlements. Line 88-90 Curriculum topics were chosen by the leaders based on the interest expressed by men in the community.

4. Results-Can you clarify why you only did the pre/post survey with one of the six groups?

Line 112 - 114 While the initial intent was to conduct the pre/post survey with all participants, travel to collect data was not possible due to the onset of the global pandemic.

Also, why only

one group debrief? It seems like a huge missed opportunity for data. Although this is recognized in the

limitation section, I think the authors should specifically address why they made this choice.

Line 107-110 All  of the 6 groups debriefed immediately after participating in the third and final meeting. However, detailed field notes were only available for one of the debriefing sessions with 12 participants due to the onset of the global pandemic.

5. Discussion-I would like to see more about the next steps for this project and what the next steps will

be?

Line 179-181 Next steps include adding additional content based on this feedback and implementing and evaluating in other settings.

Reviewer 2 Report

Comments and Suggestions for Authors

This article is based on a relevant topic of global interest and clearly presents a prevention program. The content of the prevention program is described in a clear and concise manner, as are the procedures carried out for implementation. However, there is a lack of a section in the method that addresses the methodology for evaluating impact and satisfaction with the program. Furthermore, I would like to see the Facilitator Debrief explored in greater detail, in the results section. In the Post Program Implementation Debrief section, more specifically regarding Content, I would also like to see information about the areas that participants identified as less positive. Furthermore, it appears that only 1 group of participants took the debriefing. Was it like that or is it necessary to clarify? If only 1 group did the debriefing process, why was that the case? It will be necessary to clarify.

Some more practical notes: the caption for figure 1 must be on the same page; There are some extra spaces throughout the document.

Author Response

Reviewer Comments

How and Where It was Addressed

Reviewer 2

There is a lack of a section in the method that addresses the

methodology for evaluating impact and satisfaction with the program.

A section addressing evaluation was added. Line 103-116 2.5 Evaluation

After the facilitators completed training in program implementation, they were asked to discuss their initial reaction to the curriculum including the program’s potential value and possible challenges in implementation. Detailed notes were taken during the facilitator training and of the feedback session following the training. Following program implementation, the 6 groups debriefed immediately after participating in the third and final meeting. However, detailed field notes were only available for one of the debriefing sessions with 12 participants due to the onset of the global pandemic. These participants also completed survey. A short survey was given to these participants prior to the first meeting and following the last meeting. While the initial intent was to conduct the pre/post survey with all participants, travel to collect data was not possible due to the onset of the global pandemic. The pre/post survey included 7 questions aimed to assess the difference in knowledge and awareness after the intervention but was not designed with power to determine the significance of the difference

Furthermore, I would like to see the

Facilitator Debrief explored in greater detail, in the results section.

Additional content was added to this results section.

Line 145-153 Following the training session, the facilitators were asked to openly reflect and discuss the program format as well as the  content of the curriculum. . The debrief was guided using open ended questions. Facilitators expressed confidence to implement the program and felt the practice time during the training was particularly helpful. Overall, the facilitators found the content on healthy relationship, family planning, and healthy stress reduction techniques to be important in addressing issues that can arise due to geographic and economic instability and high levels of mental health distress associated with life as a refugee. Each topic was evaluated individually during the debriefing session.

In the Post Program Implementation

Debrief section, more specifically regarding Content, I would also like to see information about the areas

that participants identified as less positive.

Additional content was added.

Line 201 - 215 Following the final meeting, one group participated in a debriefing session. Participants collectively reflected on the curriculum’s content, potential value, and areas of improvement. The participants found most value in the stress reduction techniques, feeling that stress was a significant issue for them and  some of the techniques were new to them. .For example, one of the strategies taught that was new to participants was deep breathing when you are angry. Participants enjoyed the time to practice what they learned such as communicating in calm and constructive way. While all the techniques were interesting to the participants, they felt that incorporating exercise may not be beneficial in reducing stress in their context. Participants felt that healthy communication content was important, informative, and they were eager to implement the strategies they learned. Regarding healthy lifestyle, the participants discussed difficulties with implementing some of the content given the food, employment and security challenges they faced in the settlements. However they felt the content should remain in the curriculum since it offered opportunity for discussion and problem solving.

Furthermore, it appears that only 1 group of participants took

the debriefing. Was it like that or is it necessary to clarify? If only 1 group did the debriefing process, why

was that the case? It will be necessary to clarify.

Line 107-110 All  of the 6 groups debriefed immediately after participating in the third and final meeting. However, detailed field notes were only available for one of the debriefing sessions with 12 participants due to the onset of the global pandemic.

The caption for figure 1 must be on the same page.

This has been corrected, thank you

There are some extra

spaces throughout the document.

This has been corrected, thank you